# Preliminary Development of a Brainwave Model for K1 Kickboxers Using Quantitative Electroencephalography (QEEG) with Open Eyes

**DOI:** 10.3390/ijms24108882

**Published:** 2023-05-17

**Authors:** Łukasz Rydzik, Tadeusz Ambroży, Tomasz Pałka, Wojciech Wąsacz, Michał Spieszny, Jacek Perliński, Paweł Król, Marta Kopańska

**Affiliations:** 1Institute of Sports Sciences, University of Physical Education, 31-571 Kraków, Poland; 2Department of Physiology and Biochemistry, Faculty of Physical Education and Sport, University of Physical Education, 31-571 Kraków, Poland; 3Faculty of Medical Sciences, Academy of Applied Medical and Social Sciences in Elblag, 82-300 Elblag, Poland; 4Institute of Physical Culture Studies, College of Medical Sciences, University of Rzeszow, 35-959 Rzeszów, Poland; 5Department of Pathophysiology, Institute of Medical Sciences, Medical College of Rzeszów University, 35-959 Rzeszów, Poland

**Keywords:** kickboxing, brain injury, QEEG, brain

## Abstract

K1 kickboxing fighting is characterised by high injury rates due to the low restrictions of fighting rules. In recent years, much attention has been paid to research on changes in brain function among athletes, including those in combat sports. One of the tools that are likely to help diagnose and assess brain function is quantitative electroencephalography (QEEG). Therefore, the aim of the present study was an attempt to develop a brainwave model using quantitative electroencephalography in competitive K1 kickboxers. A total of thirty-six male individuals were purposefully selected and then comparatively divided into two groups. The first group consisted of specialised K1 kickboxing athletes exhibiting a high level of sports performance (experimental group, n = 18, mean age: 29.83 ± 3.43), while the second group comprised healthy individuals not training competitively (control group, n = 18, mean age: 26.72 ± 1.77). Body composition assessment was performed in all participants before the main measurement process. Measurements were taken for kickboxers during the de-training period, after the sports competition phase. Quantitative electroencephalography of Delta, Theta, Alpha, sensimotor rhytm (SMR), Beta1 and Beta2 waves was performed using electrodes placed on nine measurement points (frontal: FzF3F4, central: CzC3C4, and parietal: PzP3P4) with open eyes. In the course of the analyses, it was found that the level of brain activity among the study population significantly differentiated the K1 formula competitors compared with the reference standards and the control group in selected measurement areas. For kickboxers, all results of the Delta amplitude activity in the area of the frontal lobe were significantly above the normative values for this wave. The highest value was recorded for the average value of the F3 electrode (left frontal lobe), exceeding the norm by 95.65%, for F4 by 74.45% and Fz by 50.6%, respectively. In addition, the Alpha wave standard value for the F4 electrode was exceeded by 14.6%. Normative values were found for the remaining wave amplitudes. Statistically significant differentiation of results, with a strong effect (d = 1.52–8.41), was shown for the activity of Delta waves of the frontal area and the central part of the parietal area (Fz,F3,F4,Cz—*p* < 0.001), Theta for the frontal area as well as the central and left parietal lobes (Fz,F3,F4—*p* < 0.001, Cz—*p* = 0.001, C3—*p* = 0.018; d = 1.05–3.18), Alpha for the frontal, parietal and occipital areas (for: Fz,F3—*p* < 0.001, F4—*p* = 0.036, Cz—*p* < 0.001, C3—*p* = 0.001, C4—*p* = 0.025, Pz—*p* = 0.010, P3—*p* < 0.001, P4—*p* = 0.038; d = 0.90–1.66), SMR for the central parietal and left occipital lobes (Cz—*p* = 0.043; d = 0.69, P3—*p* < 0.001; d = 1.62), Beta for the frontal area, occipital and central lobes and left parietal segment (Fz,F3—*p* < 0.001, F4—*p* = 0.008, Cz, C3, Pz, P3,P4—*p* < 0.001; d = 1.27–2.85) and Beta 2 for all measurement areas (Fz, F3, F4, Cz, C3, C4, Pz, P3, P4—*p* < 0.001; d = 1.90–3.35) among the study groups. Significantly higher results were shown in the kickboxer group compared to the control. In addition to problems with concentration or over-stimulation of neural structures, high Delta waves, with elevated Alpha, Theta and Beta 2 waves, can cause disorders in the limbic system and problems in the cerebral cortex.

## 1. Introduction

Kickboxing is a combat sport characterised by multiple varieties of competition [1]. The K1 rules are considered to be the kickboxing formula that allows the least limited contact. Fights under K1 rules are contested in a ring, which significantly reduces the likelihood of the opponent avoiding an attack [2,3]. K1 bouts are a special type of kickboxing in which fighters fight without limiting the impact force of the blows. The K1 rules are less strict than in other forms of kickboxing and usually allow more dynamic and aggressive fighting [4]. Due to the low number of regulatory restrictions, fighters often win their fights by a knockout [5]. A knockout is most often triggered by a hard blow to the head or body of an opponent, resulting in loss of consciousness or impaired motor function [6,7]. Scientific research on the level of technical and tactical skills of athletes competing under K1 rules has shown high effectiveness of the athletes’ attack [8,9]. This is evidenced by the large number of blows that directly reach the body of the opponent. The great number of hand and foot techniques to the head area can cause serious brain damage, such as memory disorders, mood changes, difficulty concentrating, and in extreme cases, even blindness, paralysis or death. This problem has emerged in boxing, and the consequence has been described as boxer’s encephalopathy, which is inflammation of the brain that is often caused by repeated blows to the head [10,11,12,13,14]. In numerous scientific studies, the effects of frequently repeated blows to the head in various sports have been examined, especially in combat sports. Many have confirmed that continuous and frequent impact on the head can lead to permanent brain damage and other serious health consequences [15,16,17,18,19,20]. Previous analyses in a similar area of combat sports have been conducted in the scope of spectral analysis of electroencephalographic changes following chokes in judo [21,22], as well as the neurological consequences of a knockout [23]. However, there is a lack of studies on the brainwave model in K1 kickboxing, where bouts appear to be heavier than in standard boxing matches. The only visible studies conducted in a similar scope concerned presenting kickboxing as a new cause of pituitary insufficiency [24]. Other studies concern the activity of kickboxers’ organisms using biofeedback [25]. An integrative assessment of cerebral circulation and the bioelectrical activity of kickboxers was also conducted under conditions of using corrective technologies [26]. To date, no one has addressed the issue of evaluating brain waves among kickboxing athletes, which could show how long-term participation in a high-contact sport affects changes in the brain, especially since previous analyses have demonstrated to what extent athletes receive blows directly to the head [27]. Studying the brains of kickboxing athletes is new to the field of brain research. Previous studies were mainly focused on examining the brains of athletes in sports such as football, hockey and boxing, but did not include kickboxing. Kickboxing is a high-contact sport that can lead to head injuries and the associated risk of brain damage. Therefore, analysing the brainwaves of kickboxing athletes can provide significant information regarding the long-term effects of kickboxing on the brain, as well as help prevent head injuries by better understanding kickboxing’s effects on brain function. In this way, the study may influence further research on the impact of sport on brain health, the development of more personalised training strategies and better management of head injury risk in kickboxing athletes.

Thus, the application of analyses appears to be an extremely important and significant scientific step in combating unfortunate accidents.

The EEG method is popular and widely used in the medical community, providing physicians with important diagnostic information in the field of brain functioning. Importantly, EEG is a very practical and inexpensive method of functional neuroimaging [28]. The electrical activity of the brain is recorded from the surface of the scalp, and the signals have high temporal, spatial and multi-channel resolution of the register in the long-term [29]. Analysis of recorded EEG signals allows researchers to assess the physiological state of the brain and to recognise possible neurological disorders. Electroencephalography is commonly used to diagnose epileptic seizures [30], autistic disorders [31] and also schizophrenia [29]. Quantitative electroencephalography (QEEG) is a method of studying the electrical activity of the brain using electrodes attached to the scalp [32,33]. QEEG is often used as a diagnostic tool in neuropsychology, neurology and psychiatry, as well as a tool to study the consequences of brain injuries such as boxer’s encephalopathy [13,34]. QEEG allows detailed examination of pathological changes in the brain and is often used to complement other methods such as computed tomography (CT) and magnetic resonance imaging (MRI). The test can be conducted with eyes open or closed. QEEG with open eyes records brain activity during the performance of various tasks such as looking at an object, listening to sounds or performing a cognitive task. In such tasks, the brain generates higher frequency waves associated with cognitive processes or sensory perception. The use of this technology in sports is becoming of increasing interest [35]. Analysis within the context of kickboxing can be conducted in a simplified form using nine electrodes placed on the areas most vulnerable to impact [36,37].

Taking the above premises into account, in this study, an attempt was made to develop a model of brain waves in competitive K1 kickboxers using quantitative electroencephalography (QEEG) with open eyes. The aim of the study was to assess the level of electrical brain activity and its diversity in a two-dimensional comparison of combat sports competitors (experimental group) compared to reference norms and healthy, non-training individuals (control group).

In order to achieve this objective, the following research questions were posed:Will the recognised brain activity of kickboxing athletes position itself within the reference standards for healthy people?What is the inter-group differentiation regarding the level of brain activity for individual frequency bands of the observed athletes compared to the control group?

All participants of the experimental group have many years of sports-related experience. Based on previous scientific reports, in our research, we adopted the hypothesis that due to the environmental factor in the form of long-term training and competition practice, we should expect diversification of the brain activity profile for the juxtaposed communities.

## 2. Results

In the group of kickboxers, the average values of Delta wave amplitudes were highest for electrode F3 and lowest for electrode P4. All results for the frontal lobe were well above the baseline for this wave.

High amplitude activity was observed in the frontal area (Fz, F3, F4) among kickboxers, with values significantly exceeding the norms and showing statistical significance compared to the control group. The control group exhibited higher activity in the occipital area (Pz, P3, P4) with statistical significance, although all results were within the reference norms. The same trend was observed for the peak of the skull (Cz) among kickboxers (Table 1, Figure 1).

In the results of the frequency analysis for the Theta wave, the average power values of the Theta waves in individual electrodes were highest in the Fz and F4 regions. The lowest average Theta wave power was observed for the P4 and C4 electrodes. In the comparative analysis of groups, different activity was observed for the Theta wave in the frontal area (Fz, F3, F4) and the parietal one (Cz, C3), excluding the right parietal lobe (C4), with significant statistical differences and a prevalence of results for the experimental group. The results of both groups were within the reference norms (Table 2, Figure 2).

In the experimental group, the highest value for Alpha frequency was recorded for electrode F4, while the lowest was found for F3. The comparative analysis of Alpha amplitudes revealed significant statistical differences in all measurement areas (lobes: frontal, parietal, occipital), with higher scores noted for the kickboxer group. The results of both groups were within the reference norms, except for the right frontal lobe (F4) of the kickboxer group, which exceeded the reference values (Table 3, Figure 3).

In terms of SMR frequency, the highest values were recorded for P3 and F4 in the experimental group. Statistically significant differences were found in the activity of the peak (Cz) and left occipital lobe (P3), with a higher score demonstrated for the experimental group compared to the control. The same effect in the opposite direction (significantly higher score for the control group) was observed for the right parietal lobe (C4). The SMR results of both groups were within the normal range (Table 4, Figure 4).

For Beta frequency in kickboxers, the highest values occurred in F4. Inter-group comparative analysis showed significant differences in Beta amplitude activity for all measurement areas except for the right parietal lobe (C4), with higher, yet normative results in the experimental group (Table 5, Figure 5).

In the experimental community, for the Beta 2 frequency, the highest activity was found for F4 and F3. The comparative analysis of the groups showed statistically significant differences in each measurement area, with a prevalence of normative activity in the experimental group (Table 6, Figure 6).

## 3. Discussion

The aim of the present study was to develop a brainwave model using quantitative electroencephalography in competitive K1 kickboxers. High amplitudes and percentages of Delta waves in the frontal lobe for the Fz-F3-F4 leads were observed in all athletes. The increase in the percentage of these waves is very large, reaching up to 200%. Confirmation is provided by comparative analyses with the control group, where significant differences in results were demonstrated with a strong effect. This indicates strong stimulation of the limbic system, strong emotions, a flurry of thoughts or confusion [38,39]. One can speculate that such arousal may be due to the intense emotions that accompany fighters during a kickboxing match. Stress, uncertainty, tension and the release of adrenaline are factors affecting their bodies and lead to high arousal [40,41]. Furthermore, during the fight, athletes must be fully concentrated and aware of their opponent’s moves, which can lead to a flurry of thoughts and confusion [42]. Similar conclusions have been observed in studies conducted among karate athletes [43]. However, in this study, the athletes did not fight. Therefore, the high amplitudes in the frontal lobe may have been due to the numerous blows to the head that athletes had received while competing under K1 rules [44]. The analysis of K1 fights has shown that fighters receive an average of 25 strikes in fights without headgear and as many as 52.7 in fights with headgear [27]. The training process in kickboxing is known for frequent sparring sessions. In kickboxing, athletes often throw straight punches that usually reach the forehead or guard area. The use of a block in the form of a guard protects the fighter but it transfers vibrations that may consequently induce changes to the brain [45]. Punches to the head, which are common in this combat sport, especially under K1 rules, can lead to brain damage, including that to structures responsible for generating brain waves [46,47,48,49].The hypothesis, stating that there is a difference in brain activity between kickboxers and controls due to prolonged training and competition, is supported by the results of this study. It should be noted, however, that the observed differences in brain activity, especially the high amplitudes and percentage of Delta waves in the frontal lobe, cannot be solely attributed to kickboxing training and competition. The frequent blows to the head that kickboxers receive during competition, under K1 rules in particular, can also lead to brain damage and affect brainwave generation. Therefore, it is possible that the observed differences in brain activity between kickboxers and controls may be due to a combination of factors including training, competition and head injuries.

Delta waves are some of the weakest brain waves and occur within the frequency range of 0.5–4 Hz. High Delta wave amplitudes are associated with various neurological and psychiatric disorders, such as Parkinson’s, Alzheimer’s disease, post-traumatic brain disorder and depression, as well as respiratory diseases such as obstructive pulmonary disease or sleep apnoea syndrome [50,51,52]. High Delta wave amplitudes can also occur among healthy individuals in certain physiological states such as deep sleep or a meditative state [53,54]. High Delta wave amplitudes in the frontal lobe in the group of athletes in the present study may also suggest the presence of various sleep disorders, such as sleep apnoea or limb movement disorders during sleep [55,56]. This may also be related to sleep problems resulting from constantly thinking about upcoming fights and challenges. High Delta wave amplitudes in the frontal cortex can also occur as a result of metabolic changes that are not uncommon in athletes, such as vitamin deficiency. Multiple head injuries can also induce high Delta wave amplitudes in the frontal cortex. It should be stressed that in each case, interpreting the occurrence of high Delta waves in the frontal cortex requires an analysis of the clinical context and further diagnosis (some participants in our study were instructed to undergo such examinations).

The high amplitudes of Delta waves in the frontal lobe of K1 kickboxers observed in this study raise concerns about potential neurological and psychiatric disorders, especially given the frequent hits to the head experienced by these athletes. The authors of the study suggest that these high amplitudes may be related to the emotional and cognitive functions of sport. Nonetheless, it is important to consider other possible explanations such as sleep disorders, metabolic changes or vitamin deficiencies. In addition, the clinical context of each individual athlete should be considered when interpreting these results. It should be noted that this study does not provide definitive evidence of a link between K1 kickboxing and neurological disorders. Statistical analysis also showed increased Alpha wave amplitudes in the right cerebral hemisphere of the frontal lobe (F4). In the context of a kickboxing fight, high Alpha wave values in the right hemisphere and the frontal lobe (F4) may suggest increased motor coordination and muscle control [57]. This type of above-average activity was not observed for the control group in our study. In the case of kickboxers competing under K1 rules, whose movements are fast and precise, increased activity in this region of the brain may result from intensive training, acquired knowledge and experience in fighting techniques. On the other hand, these values may also reflect the emotional arousal that accompanies sporting events, especially during a fight. An analysis of Alpha waves among elite karate athletes demonstrated lower amplitudes similar to those of non-athletes [58]. Karate is a discipline with limited regulations that prohibit punches to the head and the use of certain types of kicks to the head [59,60]. Therefore, it is worth considering whether the elevated Alpha wave values do not result from the specifics of kickboxing and frequent head strikes. It should be noted that increased amplitudes of Alpha waves in the right hemisphere of the brain in the frontal lobe (F4) may also be related to other factors such as increased attention, cognitive processing and sensory perception [61,62]. In addition, high values of Alpha waves in the right hemisphere have been associated with a positive effect and feelings of relaxation. It is therefore possible that the increased activity observed in kickboxers may be the result of both emotional arousal and increased cognitive as well as sensory processing. In future research, the mechanisms underlying increased Alpha wave activity should be investigated among kickboxers to better understand its impact on athletic performance and brain function.

Theta waves are brain waves that occur within the frequency range of approximately 4–8 Hz. The amplitude of a Theta wave is related to the height or strength of that wave. High amplitudes of Theta waves can also be associated with various emotional states such as strong arousal or the opposite, indicating a state of deep relaxation. In this group of athletes, the interviews conducted before examinations revealed variation in terms of preparation for fights or just thinking about them. Several participants reported that they had been very focused and concentrated on future fights, while others attempted to achieve a state of deep relaxation, which helps them focus on the most important effects. This is authenticated by a comparative analysis, in which significant differences in Theta wave activity (frontal and parietal regions) between combat sports athletes and the control group were found.

This may explain the elevated amplitudes of Theta and Alpha waves. Delta wave amplitudes are unfortunately more related to the large number of punches thrown to the head, especially in the frontal lobe.

Another wave with elevated amplitudes was in most cases Beta 2. This wave reflects brain activity related to the level of alertness and state of mind. High Beta 2 wave amplitudes are observed during states of increased alertness, e.g., during tasks that require attention, mental effort, stress or in situations that require quick thinking and decision making [63,64]. Beta 2 waves can also occur during motor activities. In our own research, significantly higher results were observed in kickboxers compared to the control group. In the group of athletes examined in the present study, high Beta 2 wave amplitudes can be interpreted in two ways. On the one hand, they can reflect multiple stressful situations and constant tension associated with the fights ahead, and on the other, they can be interpreted as intense focus on the challenges.

The obtained results and comparative analyses between the studied groups showed significant differences in brain wave activity in the majority of measured areas, with higher results in the kickboxer group. In the control group, the discussed parameters were lower and in line with the reference norms. This provides evidence that high-level kickboxing training affects increased activity of specific brain waves. Such a phenomenon did not occur in individuals not engaging in such high-performance sports.

### 3.1. Conclusions/Summary

In conclusion, the results show significant hyperactivity of Delta amplitudes in the frontal segment of kickboxers, which may signal a crisis associated, among others, with the strong affect, racing thoughts in the visualisation before a confrontation with a rival or the impact of blows received in the head area during a career. It is important that this type of anomaly requires further medical consultation. In addition, increased activity of the Alpha amplitude was found for the right frontal lobe section, which may indicate an above-average development of coordination abilities and broadly understood emotional arousal prior to a significant event in the form of a sports confrontation. In other areas, the kickboxers presented a normative level of brain activity, although, in most aspects, significantly higher than the control group, which may be influenced by the sports activity to which they were subjected. Systematic monitoring of athletes’ brains can provide valuable clinical information in the prevention and treatment of potential neurological disorders resulting from combat sports such as kickboxing, boxing, Muay Thai and MMA. By developing a new model of kickboxer-specific brainwaves, clinicians and therapists can use this information to plan and implement personalised therapeutic interventions to counteract any detrimental effects of training and competition on brain function.

Moreover, this approach can help identify athletes who may be at risk of developing neurological disorders such as chronic traumatic encephalopathy (CTE) and to take preventive measures as early as possible. By monitoring brain function over time, clinicians can track changes in cognitive, motor and emotional function and detect any early signs of neurological dysfunction. This proactive approach can improve athletes’ long-term health outcomes and well-being during and after their sports careers.

### 3.2. Limitation of Study

The sample size can be considered a limitation in the presented study at this stage. The study should be expanded. Additionally, quantitative electroencephalography was performed with nine cerebral cortex leads. It would be valuable to elaborate the study to include whole-head measurement of QEEG. However, in this study, we focused on the most important areas of the cerebral cortex. In addition, we did not have the opportunity to make a detailed brain map using professional software.

## 4. Materials and Methods

### 4.1. Study Design

The study was conducted on kickboxers actively competing under K1 rules during the transition after the competitive phase. Determination of the brain wave model using quantitative electroencephalography was performed using the quantitative electroencephalography (QEEG) system. All the examinations were conducted according to the Declaration of Helsinki. Each study participant provided written consent after fully reading the information provided for participants. The study was conducted on individuals who were not actively involved in sports and served as the control group in order to carry out a more precise analysis.

### 4.2. Experimental Group

The study was conducted among a group of 18 kickboxers with a high sports skill level, specialising in fighting under K1 rules. The participants were aged 29.83 ± 3.43 years. Based on Cochran’s sample size formula with a 5% of margin of error and a confidence level of 95%, the required sample size was 18 for elite athletes competing under K1 rules affiliated with the Polish Kickboxing Association. The criteria for inclusion in the study were at least 10 years of training experience, current medical examinations allowing for participation in competitions, participating in at least 5 competitions per year, a positive recommendation from the head coach and no injuries or severe knockouts during fights. The following exclusion criteria were used: short training experience, lack of active participation in competitions, injuries and heavy knockout history. All subjects were informed of the examination procedures and had not participated in sparring sessions 14 days prior to the study. Each subject recorded his/her diet on a smartphone using the Fitatu application and was instructed to refrain from consuming energy drinks and those containing caffeine or other stimulants 48 h before testing. Furthermore, prior to the study, each participant’s body composition was verified using a Tanita DC-240 MA body composition analyser (Tanita, Tokyo, Japan). Details regarding the body composition of the athletes studied are presented in Table 7.

### 4.3. Control Group

The control group consisted of 18 males aged 26.72 ± 1.77. Participants were not actively involved in sports and engaged in only low-intensity physical activity for prophylactic purposes. The inclusion criteria for the control group were age and non-participation in competitive sports, while the exclusion criteria were neurological disorders, use of psychotropic drugs and serious head injuries.

### 4.4. QEEG Procedure

QEEG (quantitative electroencephalography) is a numerical spectral analysis of the EEG record, where the data are digitally coded and statistically analysed using the Fourier transform algorithm [65,66]. Each examination of 1 person lasted about 10 min with open eyes. The wave amplitude and power for specific frequencies were analysed. Taking into account the normal values for adults, it was assumed that the lower the frequency of the waves, the lower the amplitude. Normal values were Delta waves below 20 µV, Theta below 15 µV, Alpha below 10 µV, sensorimotor rhythm (SMR), Beta 1, and Beta 2: 4–10 µV according to the standard. The EEG signal was transformed using the Cz montage and Cz electrode as the most common reference site [67] and by quantifying using Elmiko DigiTrack software (version 15, PL) (ELMIKO, Warsaw, Poland). Channels from the central lane were recorded. The study evaluated Delta, Theta, Alpha, SMR, Beta 1 and Beta 2 waves at electrodes at 9 points (frontal: FzF3F4, central: CzC3C4, and parietal: PzP3P4). The amplitude of QEEG rhythms was calculated based on medical standards using the DigiTrack apparatus. The spectrum of a signal is a representation of this signal depending on the frequency. The FFT algorithm was used, with the resulting function of f(z) = A(z) + j*F(z). In FFT analysis, the following parameters were applied: minimal signal amplitude of 0.5 µV with a minimal temporal distance between maximal values of 0.5 Hz. The analysis was performed using a computing buffer of 8.2 s (2048 assessment points, 0.12 Hz accuracy). Consequently, the set of amplitude values for each part of the frequency spectrum was obtained. The gap between single values measured in Hz defines a calculation resolution. According to the FFT algorithm, this parameter depends on signal sampling frequency and on the length of the computing buffer: r = fs/N, where r is calculation resolution, i.e., the gap between single records, fs is the signal sampling frequency, and N is the length of the computing buffer. The spectrum analysis in the FFT panel in DigiTrack showed peak-to-peak amplitudes. To ensure appropriate reliability, measurement epochs of several seconds were used [68]. The epoch length determines the frequency resolution of the Fourier transform, with a 1 s epoch providing a 1 Hz resolution (plus/minus 0.5 Hz resolution), and a 4 s epoch providing 0.25 Hz, or plus/minus 0.125 Hz resolution. The elimination of artifacts from the EEG recording was performed manually and automatically [37].

### 4.5. Methods of Statistical Analysis

Statistical analysis of the collected material was conducted via Statistica v13.3 software (TIBCO Software, California, USA). Basic descriptive statistics were calculated: arithmetic means, standard deviations, minimum, maximum, as well as the first and third quartiles. The significance of differences between the experimental and control groups was calculated using the independent samples *t*-test for independent variables. The choice of the test was determined by meeting the assumption of normal distribution, which was verified by the Shapiro–Wilk test. Additionally, effect sizes were calculated using Cohen’s d. The figure were created using Canva software, with the following normative scales for QEEG values: Delta—up to 20µV, Theta—up to 15µV, Alpha—up to 10µV, SMR—up to 6µV, Beta I—6µV, Beta 2—6µV [69,70].

## 5. Conclusions

In addition to problems with concentration or over-stimulation of neural structures, high Delta waves, with elevated Alpha, Theta and Beta 2 waves can cause disorders in the limbic system and problems in the cerebral cortex (e.g., cortical–subcortical conflict). Further research is needed to determine the exact changes in function caused by various sports activities.High amplitudes beyond the normative scale were found in the Delta, Alpha, SMR, Beta 1 and Beta 2 frequencies in the frontal lobe frequency. Therefore, it can be concluded that athletes are accompanied by an accumulation of emotions that negatively affect planning, situational assessment and coordination.The results and comparative analyses between the studied groups demonstrated significant differences regarding the activity of brain waves in the majority of the measured areas, with higher results in the kickboxer group. This suggests that the environmental influence in the form of specialised kickboxing training has an impact on the increased activity of brain waves in specific areas.Based on the presented results, it is clear that more research is needed to better understand the impact of different types of sports activities on brain function. The results suggest that specialised kickboxing training can have a significant impact on brainwave activity, highlighting the need for a more targeted and personalised approach to monitoring and treating combat sports athletes. Further research should be conducted to determine changes occurring before and after a kickboxing match and to study its long-term effects.

### Practical Implications

Systematic monitoring of athletes’ brains should be conducted to assess changes that may result from practicing martial arts or combat sports. Such action may protect them against numerous disorders and serious dysfunctions. Additionally, the development of a new model of brainwaves for kickboxers may be of high diagnostic value for planning therapeutic interventions and the possible design of new therapies to counteract the harmful effects of training and competition activity on the brain function of kickboxing K1 competitors as well as representatives of other combat sports (boxing, Muay Thai, MMA). Properly conducted QEEG diagnostics and the development of an individual therapeutic programme based on brain wave analyses can also positively affect the improvement of cognitive abilities and concentration in all groups of athletes.

## Figures and Tables

**Figure 1 ijms-24-08882-f001:**
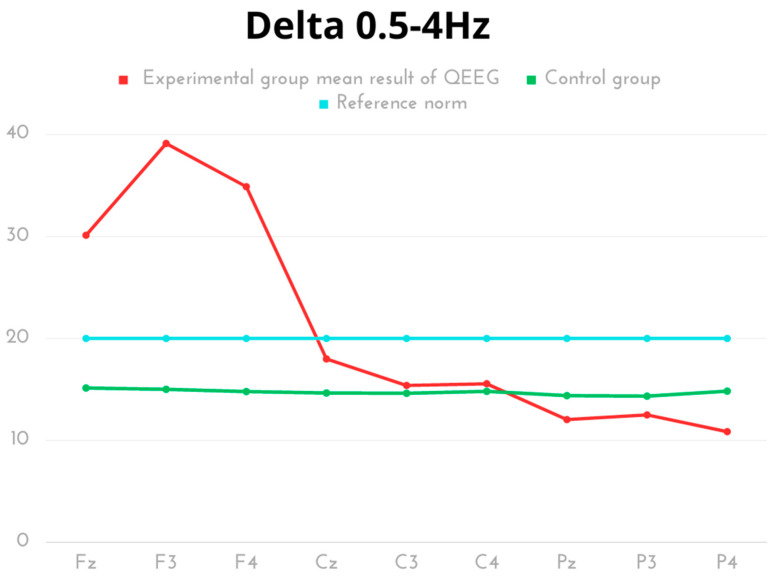
The average results of Delta frequency compared to the reference norm.

**Figure 2 ijms-24-08882-f002:**
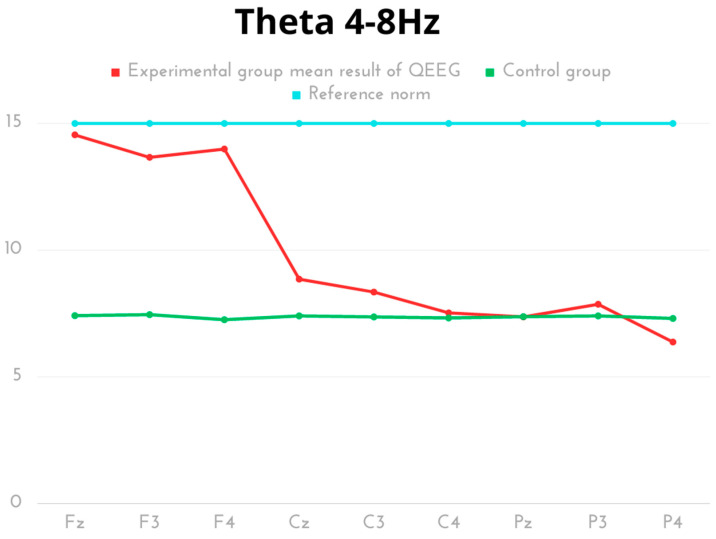
The average results of Theta frequency compared to the reference norm.

**Figure 3 ijms-24-08882-f003:**
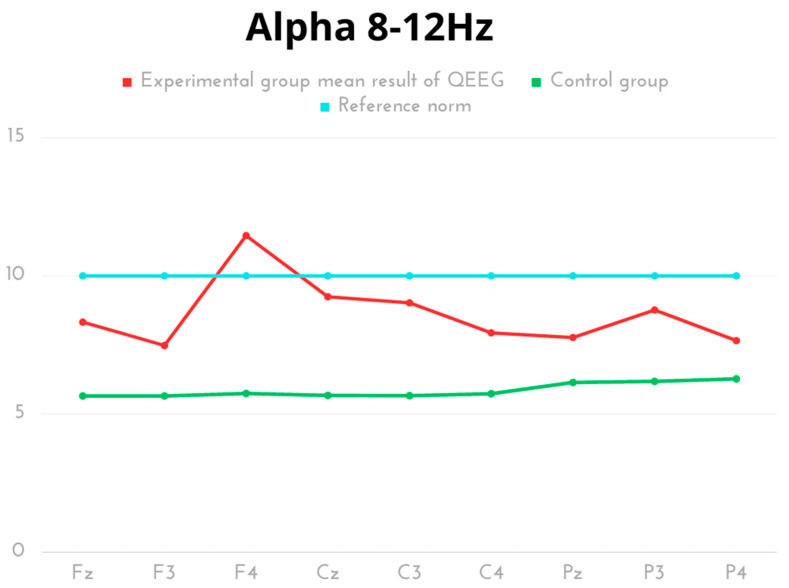
The average results of Alpha frequency compared to the reference norm.

**Figure 4 ijms-24-08882-f004:**
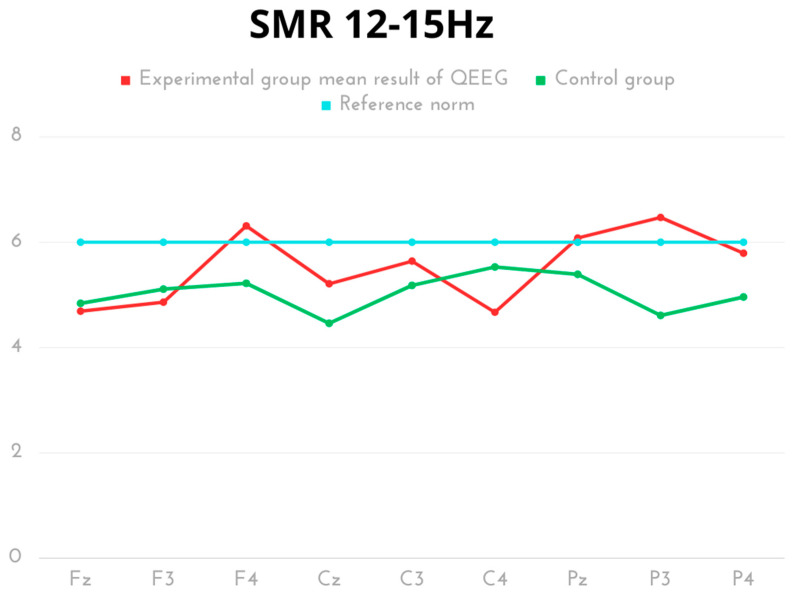
The average results of SMR frequency compared to the reference norm.

**Figure 5 ijms-24-08882-f005:**
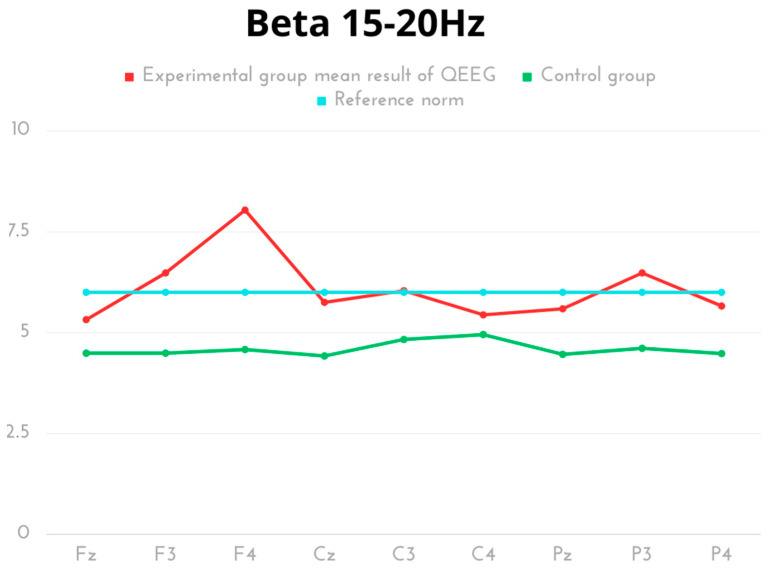
The average results of Beta frequency compared to the reference norm.

**Figure 6 ijms-24-08882-f006:**
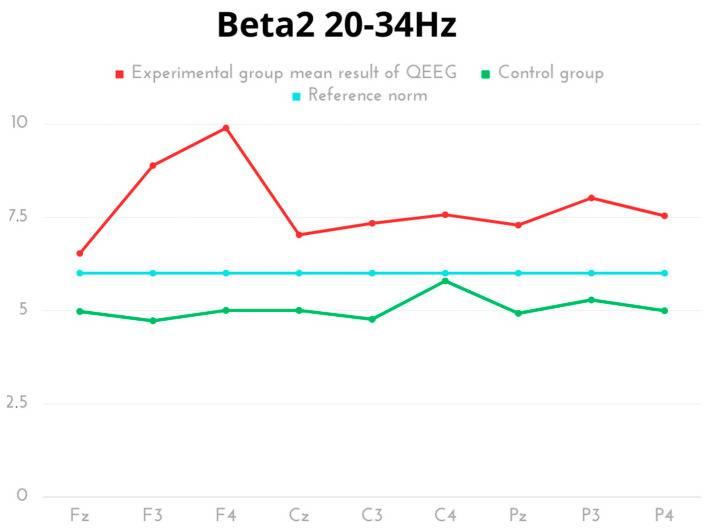
The average results of Beta2 frequency compared to the reference norm.

**Table 1 ijms-24-08882-t001:** Delta wave descriptive statistics from all 9 channels (µV).

Delta 0.5–4 Hz	Mean	SD	Min	Max	Q1	Q3	*p*	Cohen’s d
Experimental Fz	30.12	6.19	21.37	40.65	25.60	34.09	** *p* ** **< 0.001**	**4.23**
Control Fz	15.14	0.89	14.30	16.98	14.56	15.43		
Experimental F3	39.13	6.16	33.53	50.56	34.89	43.22	** *p* ** **< 0.001**	**7.03**
Control F3	15.01	0.70	14.23	15.98	14.43	15.56		
Experimental F4	34.89	3.81	29.59	38.74	30.62	38.64	** *p* ** **< 0.001**	**8.41**
Control F4	14.79	0.97	13.28	15.92	14.63	15.69		
Experimental Cz	17.99	3.88	13.02	24.40	15.36	20.36	** *p* ** **< 0.001**	**1.52**
Control Cz	14.65	0.52	14.21	15.89	14.43	14.56		
Experimental C3	15.39	3.02	12.85	20.63	13.32	18.21	*p* = 0.288	0.44
Control C3	14.62	0.46	14.23	15.90	14.43	14.56		
Experimental C4	15.55	4.10	11.63	23.24	12.05	17.72	*p* = 0.460	0.29
Control C4	14.81	1.01	13.25	15.98	14.63	15.88		
Experimental Pz	12.04	1.61	8.79	14.4	10.87	12.82	** *p* ** **< 0.001**	**2.04**
Control Pz	14.39	0.69	13.43	15.12	13.63	14.96		
Experimental P3	12.50	1.56	10.04	14.40	11.46	14.26	** *p* ** **< 0.001**	**1.66**
Control P3	14.34	0.66	13.39	15.29	13.63	14.76		
Experimental P4	10.85	1.67	8.66	12.82	8.79	12.47	** *p* ** **< 0.001**	**3.10**
Control P4	14.83	0.90	13.48	15.95	14.63	15.73		

Statically significant values are shown in bold.

**Table 2 ijms-24-08882-t002:** Theta wave descriptive statistics from all 9 channels (µV).

Theta 4–8 Hz	Mean	SD	Min	Max	Q1	Q3	*p*	Cohen’s d
Experimental Fz	14.55	5.30	9.19	27.55	11.77	14.91	*p* < 0.001	2.62
Control Fz	7.42	0.14	7.18	7.62	7.27	7.54		
Experimental F3	13.66	4.08	9.88	22.19	11.78	13.70	*p* < 0.001	2.97
Control F3	7.46	0.10	7.21	7.54	7.42	7.54		
Experimental F4	13.99	3.87	9.90	22.37	11.91	14.72	*p* < 0.001	3.18
Control F4	7.26	0.36	6.54	7.89	6.98	7.47		
Experimental Cz	8.86	1.70	6.69	11.67	7.38	10.10	*p* = 0.001	1.57
Control Cz	7.41	0.15	7.08	7.58	7.29	7.54		
Experimental C3	8.35	1.67	6.59	11.10	6.77	9.07	*p* = 0.018	1.05
Control C3	7.37	0.19	7.06	7.59	7.19	7.54		
Experimental C4	7.53	1.78	5.78	11.11	6.58	8.08	*p* = 0.628	0.18
Control C4	7.33	0.43	6.54	7.83	6.95	7.65		
Experimental Pz	7.37	1.63	5.14	10.07	6.19	8.16	*p* = 0.976	0.01
Control Pz	7.38	0.18	7.08	7.61	7.23	7.54		
Experimental P3	7.87	1.65	6.01	10.07	6.54	9.84	*p* = 0.241	0.51
Control P3	7.41	0.15	7.04	7.54	7.33	7.54		
Experimental P4	6.38	1.33	5.08	8.15	5.11	7.87	*p* = 0.051	1.04
Control P4	7.31	0.46	6.54	7.89	6.86	7.65		

**Table 3 ijms-24-08882-t003:** Alpha wave descriptive statistics from all 9 channels (µV).

Alpha 8–12 Hz	Mean	SD	Min	Max	Q1	Q3	*p*	Cohen’s d
Experimental Fz	8.32	2.87	6.31	14.34	6.42	8.62	*p* < 0.001	1.65
Control Fz	5.64	0.38	5.05	6.18	5.43	5.93		
Experimental F3	7.47	1.99	5.45	11.56	6.51	7.58	*p* < 0.001	1.54
Control F3	5.64	0.38	5.02	6.13	5.43	5.93		
Experimental F4	11.46	11.13	4.73	35.37	5.58	10.12	*p* = 0.036	1.00
Control F4	5.73	0.33	5.29	6.17	5.39	6.03		
Experimental Cz	9.24	3.95	5.58	16.82	5.66	10.70	*p* < 0.001	1.66
Control Cz	5.66	0.37	5.08	6.21	5.43	5.93		
Experimental C3	9.02	4.10	5.34	17.37	5.71	8.97	*p* = 0.001	1.52
Control C3	5.65	0.33	5.11	6.16	5.43	5.93		
Experimental C4	7.93	3.98	4.57	16.18	4.88	7.84	*p* = 0.025	1.03
Control C4	5.72	0.31	5.29	6.17	5.40	5.97		
Experimental Pz	7.76	2.53	4.68	12.07	5.62	9.64	*p* = 0.010	1.20
Control Pz	6.13	0.18	5.93	6.43	5.95	6.24		
Experimental P3	8.76	2.99	4.68	13.09	6.18	11.69	*p* < 0.001	1.51
Control P3	6.17	0.45	5.56	6.79	5.79	6.65		
Experimental P4	7.65	2.72	4.60	11.98	4.75	9.94	*p* = 0.038	0.90
Control P4	6.26	0.36	5.79	6.83	5.99	6.61		

**Table 4 ijms-24-08882-t004:** SMR wave descriptive statistics from all 9 channels (µV).

SMR 12–15 Hz	Mean	SD	Min	Max	Q1	Q3	*p*	Cohen’s d
Experimental Fz	4.69	0.81	3.74	6.14	4.00	5.08	*p* = 0.606	0.17
Control Fz	4.84	0.93	3.73	5.92	3.73	5.78		
Experimental F3	4.86	0.87	3.67	6.11	3.83	5.31	*p* = 0.297	0.38
Control F3	5.11	0.46	4.44	5.75	4.98	5.57		
Experimental F4	6.31	4.37	3.12	15.64	3.58	5.25	*p* = 0.300	0.44
Control F4	5.22	0.57	4.58	5.91	4.73	5.82		
Experimental Cz	5.21	1.12	3.68	7.02	4.16	5.87	*p* = 0.043	0.69
Control Cz	4.46	1.04	3.12	5.73	3.92	5.73		
Experimental C3	5.64	1.29	3.79	7.58	4.40	6.34	*p* = 0.165	0.51
Control C3	5.18	0.51	4.52	5.93	4.98	5.70		
Experimental C4	4.67	1.22	3.00	6.60	3.48	5.41	*p* = 0.009	1.02
Control C4	5.53	0.47	4.55	5.94	5.47	5.87		
Experimental Pz	6.08	1.84	3.52	9.18	4.55	7.28	*p* = 0.129	0.61
Control Pz	5.39	0.42	4.61	5.92	5.23	5.81		
Experimental P3	6.47	1.81	4.39	8.80	4.45	8.47	*p* < 0.001	1.62
Control P3	4.61	0.48	3.93	5.01	4.06	4.98		
Experimental P4	5.79	2.06	3.52	9.18	3.58	7.38	*p* = 0.116	0.59
Control P4	4.96	0.77	3.92	5.73	4.09	5.73		

**Table 5 ijms-24-08882-t005:** BETA wave descriptive statistics from all 9 channels (µV).

Beta 15–20 Hz	Mean	SD	Min	Max	Q1	Q3	*p*	Cohen’s d
Experimental Fz	5.32	0.67	4.50	6.59	5.03	5.63	*p* < 0.001	1.71
Control Fz	4.49	0.30	4.01	4.82	4.46	4.82		
Experimental F3	6.48	1.60	4.98	9.77	5.60	6.77	*p* < 0.001	2.09
Control F3	4.49	0.30	4.01	4.82	4.46	4.82		
Experimental F4	8.04	5.25	4.09	19.30	5.58	6.69	*p* = 0.008	1.27
Control F4	4.58	0.18	4.39	4.82	4.44	4.82		
Experimental Cz	5.75	0.94	4.810	7.59	5.20	6.19	*p* < 0.001	2.40
Control Cz	4.42	0.17	4.12	4.56	4.39	4.56		
Experimental C3	6.04	1.06	4.80	7.83	5.38	6.93	*p* < 0.001	1.41
Control C3	4.83	0.66	4.32	5.98	4.32	4.87		
Experimental C4	5.44	0.92	4.21	6.88	4.82	6.27	*p* = 0.073	0.62
Control C4	4.95	0.65	4.36	5.82	4.41	5.82		
Experimental Pz	5.59	1.056	4.21	7.51	4.53	6.33	*p* < 0.001	1.66
Control Pz	4.46	0.30	4.01	4.78	4.33	4.78		
Experimental P3	6.48	1.04	4.91	7.96	5.89	7.44	*p* < 0.001	2.85
Control P3	4.61	0.27	4.38	5.01	4.38	4.85		
Experimental P4	5.66	1.10	4.43	7.51	4.44	6.35	*p* < 0.001	1.66
Control P4	4.48	0.32	4.08	4.88	4.28	4.88		

**Table 6 ijms-24-08882-t006:** Beta 2 wave descriptive statistics from all 9 channels (µV).

Beta 2 20–35 Hz	Mean	SD	Min	Max	Q1	Q3	*p*	Cohen’s d
Experimental Fz	6.53	0.59	5.66	7.96	6.16	6.89	*p* < 0.001	2.15
Control Fz	4.97	0.86	4.12	5.96	4.12	5.89		
Experimental F3	8.89	2.72	6.06	14.67	7.31	9.41	*p* < 0.001	2.82
Control F3	4.72	0.24	4.51	5.12	4.51	4.74		
Experimental F4	9.90	3.29	5.21	16.17	7.08	12.56	*p* < 0.001	2.61
Control F4	5.00	0.47	4.34	5.51	4.58	5.51		
Experimental Cz	7.03	0.76	6.18	9.25	6.25	7.49	*p* < 0.001	2.43
Control Cz	5.00	0.91	4.07	5.96	4.07	5.89		
Experimental C3	7.34	1.02	6.14	9.82	6.40	7.64	*p* < 0.001	3.35
Control C3	4.76	0.52	4.32	5.69	4.46	4.65		
Experimental C4	7.57	0.96	6.38	9.28	6.69	7.89	*p* < 0.001	1.90
Control C4	5.79	0.91	4.82	6.91	4.95	6.91		
Experimental Pz	7.29	0.83	6.18	9.28	6.69	7.73	*p* < 0.001	2.72
Control Pz	4.92	0.91	4.01	5.89	4.01	5.88		
Experimental P3	8.02	0.79	7.08	9.29	7.10	8.46	*p* < 0.001	3.08
Control P3	5.28	0.99	4.17	6.51	4.34	6.51		
Experimental P4	7.54	1.09	5.85	8.83	7.14	8.60	*p* < 0.001	3.00
Control P4	4.99	0.61	4.22	5.54	4.32	5.54		

**Table 7 ijms-24-08882-t007:** Body composition in the study group.

Parameter	Mean	SD	Min	Max	Q1	Q3
Body mass [kg]	81.16	8.43	69.60	91.80	73.55	89.30
Body height [cm]	178.50	4.87	170.0	185.00	175.50	182.00
BMI	25.30	2.33	21.50	28.40	23.55	27.05
BF [%]	16.80	5.07	7.10	24.60	13.50	21.30
FFM [kg]	13.94	5.30	5.20	22.00	9.85	19.40
LMB [kg]	67.21	4.51	59.10	74.60	64.60	70.20
TBW [kg]	46.71	3.18	42.80	52.40	44.00	49.40
SMM [kg]	63.87	4.32	56.10	70.90	61.35	66.75
BMD [kg]	3.34	0.19	3.00	3.70	3.25	3.45
Metabolic age [years]	22.91	8.72	13.00	38.00	15.50	32.00

BMI—body mass index, BF—body fat percentage, FFM—fat-free mass, LMB—lean body mass, TBW—total body water, SMM—skeletal muscle mass, BMD—bone mineral density SD—standard deviation, Q1— first quartile, Q3—third quartile.

## Data Availability

All data are included in the manuscript.

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
