# Peer review of "Preliminary Development of a Brainwave Model for K1 Kickboxers Using Quantitative Electroencephalography (QEEG) with Open Eyes"

_ijms, 2023, doi:10.3390/ijms24108882_

Round 1

Reviewer 1 Report (Previous Reviewer 1)

Accept in present form

Author Response

Dear Reviewer

Thank you.

Your Sincerly, 

Łukasz Rydzik 

Reviewer 2 Report (New Reviewer)

After carefully reviewing this paper, I have determined that it lacks novelty and is not suitable for publication. There are several concerns that I have identified:

The paper fails to clearly demonstrate its novelty.

The method used to analyze EEG signals is overly simplistic.

The abstract is poorly written as it fails to provide sufficient detail about the methodology and results.

The related works section is inadequate.

The dataset section is not clearly explained and would benefit from the inclusion of EEG signal plots.

The paper lacks both a discussion and limitations of the study section.

The paper fails to meet the standards necessary for publication. It is recommended that the authors revise the paper to address the aforementioned concerns.

Author Response

Dear Reviewer,

Thank you very much for your time and valuable comments, which all have been considered and incorporated. The detailed list of responses is given below. We hope that the modifications and explanation will be acceptable for you.

Yours sincerely,

Rydzik, corresponding author

After carefully reviewing this paper, I have determined that it lacks novelty and is not suitable for publication. There are several concerns that I have identified:

A: We hope that the modifications made will be acceptable to you.

The paper fails to clearly demonstrate its novelty.

A: We have added additional information regarding our new study. I would like to point out that such studies have not been conducted before. Searching for "EEG kickboxing" on PUBMED yields zero results, and a similar situation occurs with QEEG kickboxing. It seems that no one has conducted such analyses before, and we are the only ones in this field. Additionally, we are studying the K1 style, which is the most contact-intensive among all kickboxing styles. We have conducted pilot studies in this area. To further develop the scientific problem, we have added the results of a control group to the study. This comparison shows how kickboxing athletes differ from people not involved in sports. The small number of participants in the study is due to the elitism of this sport and style; at the best K1 event in our country, there are only 40 athletes (though of world-class caliber). Therefore, our group consists of excellent athletes, but there are very few of them.

The method used to analyze EEG signals is overly simplistic.

A: Thank you for your comment. I would like to clarify that in our study, we used QEEG, not EEG. The research protocol was derived from other scientific studies. From a kickboxing perspective, the use of 9 leads seems reasonable. Analysis shows that athletes most frequently receive blows to the front of the head and its sides. Therefore, this seems appropriate. Additionally, our results showed significant changes, indicating that the method is valid. Below, I am sending studies that were conducted using a similar method.

https://www.mdpi.com/1424-8220/22/17/6606

The abstract is poorly written as it fails to provide sufficient detail about the methodology and results.

A: Thank you for your comment. The abstract has been rewritten

The related works section is inadequate.

A: We have made changes in this regard.

The dataset section is not clearly explained and would benefit from the inclusion of EEG signal plots.

A:We have added a control group for better visualization and have included the unit of measurement. The positive opinion of your reviewer and the associated possible publication of this manuscript will give us a chance to bypass one of our research limitations. By obtaining a large grant, we will be able to purchase special equipment and software. All this will enable us to continue our research.

The paper lacks both a discussion and limitations of the study section.

A: The discussion has been modified , also the limitations of the study have been extended

The paper fails to meet the standards necessary for publication. It is recommended that the authors revise the paper to address the aforementioned concerns.

Round 2

Reviewer 2 Report (New Reviewer)

1. The abstract is not organized well. Please provide more details of proposed model and results in this section.

2. Please explain more about EEG signals for diagnosis of brain disorders. I recommended some references that you can used some references such as https://doi.org/10.1016/j.bspc.2021.103417 and  https://doi.org/10.1007/978-3-031-06242-1_7  for this section.

3. The research question(s) need to appear stronger and clearer.

4. Please clarify your initial hypothesis.

5. In discussions you need to critically discuss your work/results against your hypothesis.

6. Identify the main findings and justify the novelty and contribution of the work.

7. A recap of all the relevant parameters with their meaning should be added to help the reader.

8. Please highlight the clinical significance of your findings.

9. Please add a section about "limitation of study".

10. In the Conclusion section, please explain more about future works. This section requires further discussion.

11. English language is acceptable in general, but there are some errors that should be corrected.

Author Response

Dear Reviewer,

Thank you very much for your time and valuable comments, which all have been considered and incorporated. The detailed list of responses is given below. We hope that the modifications and explanation will be acceptable for you.

Yours sincerely,

Rydzik, corresponding author

  1. The abstract is not organized well. Please provide more details of proposed model and results in this section.

A: The results section of the abstract has been rewritten with a detailed description of the model. We hope that the changes will be acceptable to you.

  1. Please explain more about EEG signals for diagnosis of brain disorders. I recommended some references that you can used some references such as https://doi.org/10.1016/j.bspc.2021.103417 and  https://doi.org/10.1007/978-3-031-06242-1_7  for this section.

A: Thank you for the hint. We have added more information in the introduction section considering the proposed articles

  1. The research question(s) need to appear stronger and clearer.

A: We have added newly developed research questions

  1. Please clarify your initial hypothesis.

A: Added hypothesis , which was placed under the research questions

  1. In discussions you need to critically discuss your work/results against your hypothesis.

A: The discussion was expanded to include a critical approach and verification of the hypothesis

  1. Identify the main findings and justify the novelty and contribution of the work.

A: Added information in introduction and discussion

  1. A recap of all the relevant parameters with their meaning should be added to help the reader.

A: Added a summary in the form of a subsection for discussion

  1. Please highlight the clinical significance of your findings.

A: Added information in practical implication, discussion and introduction

  1. Please add a section about "limitation of study".

A: Added section "limitation of study"

  1. In the Conclusion section, please explain more about future works. This section requires further discussion.

A: Added information and formulated a new conclusion

  1. English language is acceptable in general, but there are some errors that should be corrected.

A: The entire manuscript has been checked by a native speaker and confirmed the quality of the language with the attached certificate

Round 3

Reviewer 2 Report (New Reviewer)

Thanks for your excellent work. This version of the paper can be accepted for publication. 

This manuscript is a resubmission of an earlier submission. The following is a list of the peer review reports and author responses from that submission.

Round 1

Reviewer 1 Report

In this paper, Rydzik et al. present evidence an attempt to develop a brain wave model using quantitative electroencephalography with eyes open in K1 kickboxers. The results of the study have been showing the Quantitative electroencephalography (qEEG) is a method that measures the electrical activity of the brain, which can be used to develop models of brain waves. K1 kickboxing is a form of combat sport that requires intense physical and mental training, which can have an impact on brain activity. Therefore, studying the brain waves of K1 kickboxers during rest can provide insights into the impact of the sport on their brain activity. The manuscript has some concerns. The reviewer makes the following suggestions:

1. qEEG should be conducted while the participants are seated with their eyes open in a quiet and dimly lit room. The recording should be conducted using a 19-channel EEG system with a sampling rate of at least 500 Hz.

2. The recorded EEG data should be processed using a standard EEG analysis software. The data should be pre-processed by removing any artifacts such as eye blinks, muscle activity, and other sources of noise.

3. The next step involves extracting features from the pre-processed EEG data. These features can include power spectral density (PSD), coherence, phase, and other measures of spectral power and connectivity.

4. The final step involves developing a brain wave model using the significant features identified in the statistical analysis. This model can be used to predict brain wave patterns in K1 kickboxers based on their age, training experience, and other relevant factors.

5. The discussion should be improved by correlating these results with the literature.

6. The manuscript is not linked to current conversations in the journal.

7. Most importantly, there is too little data to support the study.

Reviewer 2 Report

The paper presented an interesting application.However, more technical details can imporve the paper.

Lack 2023 references.

Please explain with computing technical details.

Please give formal proof of claims in the conslusion.

Please explain complexity of relevant solutions.

Please explain validity and generalisability of relevant solutions.

Please explain originality and advantages of own solutions comparing with existing solutions.

Explain the paper's solution's limitation.

Reviewer 3 Report

The authors have come across with good attempt to carry out the research in this direction but the manuscript needs improvement in the scientific approach.

Major Comment is as follows:

1.      The study lacks novelty. Authors should carry out this study with an innovative technique and with a good conceptual paradigm beneficial for public health.
